# EEC: Learning to Encode and Regenerate Images for Continual Learning

**Ali Ayub, Alan R. Wagner**
The Pennsylvania State University
State College, PA, USA, 16803
{aja5755,alan.r.wagner}@psu.edu.edu

## Abstract

The two main impediments to continual learning are catastrophic forgetting and memory limitations on the storage of data. To cope with these challenges, we propose a novel, cognitively-inspired approach which trains autoencoders with Neural Style Transfer to encode and store images. During training on a new task, reconstructed images from encoded episodes are replayed in order to avoid catastrophic forgetting. The loss function for the reconstructed images is weighted to reduce its effect during classifier training to cope with image degradation. When the system runs out of memory the encoded episodes are converted into centroids and covariance matrices, which are used to generate pseudo-images during classifier training, keeping classifier performance stable while using less memory. Our approach increases classification accuracy by **13-17%** over state-of-the-art methods on benchmark datasets, while requiring **78%** less storage space.[1]

## 1 Introduction

Humans continue to learn new concepts over their lifetime without the need to relearn most previous concepts. Modern machine learning systems, however, require the complete training data to be available at one time (batch learning) (Girshick, 2015). In this paper, we consider the problem of continual learning from the class-incremental perspective. Class-incremental systems are required to learn from a stream of data belonging to different classes and are evaluated in a single-headed evaluation (Chaudhry et al., 2018). In single-headed evaluation, the model is evaluated on all classes observed so far without any information indicating which class is being observed.

Creating highly accurate class-incremental learning systems is a challenging problem. One simple way to create a class-incremental learner is by training the model on the data of the new classes, without revisiting the old classes. However, this causes the model to forget the previously learned classes and the overall classification accuracy decreases, a phenomenon known as *catastrophic forgetting* (Kirkpatrick et al., 2017). Most existing class-incremental learning methods avoid this problem by storing a portion of the training samples from the earlier learned classes and retraining the model (often a neural network) on a mixture of the stored data and new data containing new classes (Rebuffi et al., 2017; Hou et al., 2019). Storing real samples of the previous classes, however, leads to several issues. First, as pointed out by Wu et al. (2018b), storing real samples exhausts memory capacity and limits performance for real-world applications. Second, storing real samples introduces privacy and security issues (Wu et al., 2018b). Third, storing real samples is not biologically inspired, i.e. humans do not need to relearn previously known classes.

This paper explores the "strict" class-incremental learning problem in which the model is not allowed to store any real samples of the previously learned classes. The strict class-incremental learning problem is more akin to realistic learning scenarios such as a home service robot that must learn continually with limited on-board memory. This problem has been previously addressed using generative models such as autoencoders (Kemker & Kanan, 2018) or Generative Adversarial Networks (GANs) (Ostapenko et al., 2019). Most approaches for strict class-incremental learning

---

[1]A preliminary version of this work was presented at ICML 2020 Workshop on Lifelong Machine Learning (Ayub & Wagner, 2020c).

use GANs to generate samples reflecting old class data, because GANs generate sharp, fine-grained images (Ostapenko et al., 2019). The downside of GANs, however, is that they tend to generate images which do not belong to any of the learned classes, hurting classification performance. Autoencoders, on the other hand, always generate images that relate to the learned classes, but tend to produce blurry images that are also not good for classification.

To cope with these issues, we propose a novel, cognitively-inspired approach termed Encoding Episodes as Concepts (EEC) for continual learning, which utilizes convolutional autoencoders to generate previously learned class data. Inspired by models of the hippocampus (Renoult et al., 2015), we use autoencoders to create compressed embeddings (encoded episodes) of real images and store them in memory. To avoid the generation of blurry images, we borrow ideas from the Neural Style Transfer (NST) algorithm proposed by Gatys et al. (2016) to train the autoencoders. For efficient memory management, we use the notion of *memory integration*, from hippocampal and neocortical concept learning (Mack et al., 2018), to combine similar episodes into centroids and covariance matrices eliminating the need to store real data.

This paper contributes: 1) an autoencoder based approach to strict class-incremental learning which uses Neural Style Transfer to produce quality samples reflecting old class data (Sec. 3.1); 2) a cognitively-inspired memory management technique that combines similar samples into a centroid/covariance representation, drastically reducing the memory required (Sec. 3.2); 3) a data filtering and a loss weighting technique to manage image degradation of old classes during classifier training (Sec. 3.3). We further show that EEC outperforms state-of-the-art (SOTA) approaches on benchmark datasets by significant margins while also using far less memory.

## 2  RELATED WORK

Most recent approaches to class-incremental learning store a portion of the real images belonging to the old classes to avoid catastrophic forgetting. Rebuffi et al. (2017) (iCaRL) store old class images and utilize knowledge distillation (Hinton et al., 2015) for representation learning and the nearest class mean (NCM) classifier for classification of the old and new classes. Knowledge distillation uses a loss term to force the labels of the images of previous classes to remain the same when learning new classes. Castro et al. (2018) (EEIL) improves iCaRL with an end-to-end learning approach. Wu et al. (2019) also stores real images and uses a bias correction layer to avoid any bias toward the new classes.

To avoid storing old class images, some approaches store features from the last fully-connected layer of the neural networks (Xiang et al., 2019; Hayes & Kanan, 2020; Ayub & Wagner, 2020b;d). These approaches, however, use a network pretrained on ImageNet to extract features, which gives them an unfair advantage over other approaches. Because of their reliance on a pretrained network, these approaches cannot be applied in situations when new data differs drastically from ImageNet (Russakovsky et al., 2015).

These difficulties have forced researchers to consider using generative networks. Methods employing generative networks tend to model previous class statistics and regenerate images belonging to the old classes while attempting to learn new classes. Both Shin et al. (2017) and Wu et al. (2018a) use generative replay where the generator is trained on a mixture of generated old class images and real images from the new classes. This approach, however, causes images belonging to classes learned in earlier increments to start to semantically drift, i.e. the quality of images degrades because of the repeated training on synthesized images. Ostapenko et al. (2019) avoids semantic drift by training the GAN only once on the data of each class. Catastrophic forgetting is avoided by applying elastic weight consolidation (Kirkpatrick et al., 2017), in which changes in important weights needed for old classes are avoided when learning new classes. They also grow their network when it runs out of memory while learning new classes, which can be difficult to apply in situations with restricted memory. One major issue with GAN based approaches is that GANs tend to generate images that do not belong to any of the learned classes which decreases classification accuracy. For these reasons, most approaches only perform well on simpler datasets such as MNIST (LeChun, 1998) but perform poorly on complex datasets such as ImageNet. Conditional GAN can be used to mitigate the problem of images belonging to none of the classes as done by Ostapenko et al. (2019), however the performance is still poor on complex datasets such as ImageNet-50 (see Table 1 and

Table 2). We avoid the problem of generating images that do not belong to any learned class by training autoencoders instead of GANs.

Comparatively little work has focused on using autoencoders to generate samples because the images generated by autoencoders are blurry, limiting their usefulness for classifier training. Hattori (2014) uses autoencoders on binary pixel images and Kemker & Kanan (2018) (FearNet) uses a network pre-trained on ImageNet to extract feature embeddings for images, applying the autoencoder to the feature embeddings. Neither of these approaches are scalable to RGB images. Moreover, the use of a pre-trained network to extract features gives FearNet an unfair advantage over other approaches.

## 3 ENCODING EPISODES AS CONCEPTS (EEC)

Following the notation of Chaudhry et al. (2018), we consider $S_t = \{(x_i^t, y_i^t)\}_{i=1}^{n^t}$ to be the set of samples $x_i \in \mathcal{X}$ and their ground truth labels $y_i^t$ belonging to task $t$. In a class-incremental setup, $S_t$ can contain one or multiple classes and data for different tasks is available to the model in different increments. In each increment, the model is evaluated on all the classes seen so far.

Our formal presentation of continual learning follows Ostapenko et al. (2019), where a task solver model (classifier for class-incremental learning) $D$ has to update its parameters $\theta_D$ on the data of task $t$ in an increment such that it performs equally well on all the $t - 1$ previous tasks seen so far. Data for the $t - 1$ tasks is not available when the model is learning task $t$. The subsections below present our approach.

### 3.1 AUTOENCODER TRAINING WITH NEURAL STYLE TRANSFER

An autoencoder is a neural network that is trained to compress and then reconstruct the input (Goodfellow et al., 2016), formally $f_r : \mathcal{X} \to \mathcal{X}$. The network consists of an encoder that compresses the input into a lower dimensional feature space (termed as the encoded episode in this paper), $g_{enc} : \mathcal{X} \to \mathcal{F}$ and a decoder that reconstructs the input from the feature embedding, $g_{dec} : \mathcal{F} \to \mathcal{X}$. Formally, for a given input $x \in \mathcal{X}$, the reconstruction pipeline $f_r$ is defined as: $f_r(x) = (g_{dec} \circ g_{enc})(x)$. The parameters $\theta_r$ of the network are usually optimized using an $l_2$ loss $(\mathcal{L}_r)$ between the inputs and the reconstructions:

$$\mathcal{L}_r = ||x - f_r(x)||_2 \tag{1}$$

Although autoencoders are suitable for dimensionality reduction for complex, high-dimensional data like RGB images, the reconstructed images lose the high frequency components necessary for correct classification. To tackle this problem, we train autoencoders using some of the ideas that underline Neural Style Transfer (NST). NST uses a pre-trained CNN to transfer the style of one image to another. The process takes three images, an input image, a content image and a style image and alters the input image such that it has the content image's content and the artistic style of the style image. The three images are passed through the pre-trained CNN generating convolutional feature maps (usually from the last convolutional layer) and $l_2$ distances between the feature maps of the input image and content image (content loss) and style image (style loss) are calculated. These losses are then used to update the input image.

Intuitively, our intent here is to create reconstructed images that are similar to the real images (in the pixel and convolutional space), thereby improving classification accuracy. Hence, we only utilize the idea of content transfer from the NST algorithm, where the input image is the image reconstructed by the autoencoder and content image is the real image corresponding to the reconstructed image. The classifier model, $D$, is used to generate convolutional feature maps for the NST, since it is already trained on real data for the classes in the increment $t$. In contrast to the traditional NST algorithm, we use the content loss ($\mathcal{L}_{cont}$) to train the autoencoder, rather than updating the input image directly. Formally, let $f_c : \mathcal{X} \to \mathcal{F}_c$ be the classifier pipeline that converts input images into convolutional features. For an input image, $x_i^t$ of task $t$, the content loss is:

$$\mathcal{L}_{cont} = ||f_c(x_i^t) - f_c(f_r(x_i^t))||_2 \tag{2}$$

The autoencoder parameters are optimized using a combination of reconstruction and content losses:

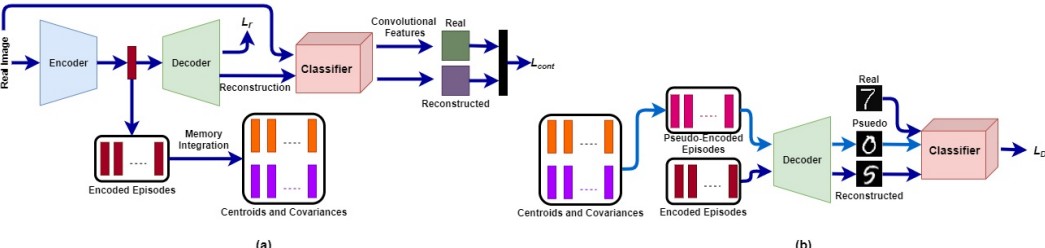

Figure 1: Complete architecture of EEC. (a) For each new task, a convolutional autoencoder is trained on real images and a combination of reconstruction loss $\mathcal{L}_r$ and content loss $\mathcal{L}_{cont}$. The encoded episodes are stored in memory and converted into centroids and covariance matrices when the system runs out of memory. (b) The classifier is trained on a combination of real, reconstructed and pseudo-images in each increment.

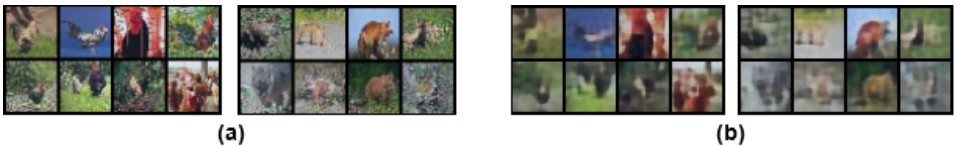

Figure 2: Reconstructed images generated by autoencoders trained (a) using NST and (b) not using NST .

$$\mathcal{L} = (1 - \lambda)\mathcal{L}_r + \lambda\mathcal{L}_{cont} \qquad (3)$$

where, $\lambda$ is a hyperparamter that controls the contribution of each loss term towards the complete loss. During autoencoder training, classifier $D$ acts as a fixed feature extractor and its parameters are not updated. This portion of the complete procedure is depicted in Figure 1 (a).

To provide an illustration of our approach, we perform an experiment with ImageNet-50 dataset. We trained one autoencoder on 10 classes from ImageNet-50 with NST and one without NST. Figure 2 depicts the reconstructed images by the two autoencoders. Note that the images generated by the autoencoder trained without using NST are blurry. In contrast, the autoencoder trained using NST creates images with fine-grained details which improves the classification accuracy.

### 3.2 MEMORY INTEGRATION

As mentioned in the introduction, continual learning also presents issues associated with the storage of data in memory. For EEC, for each new task $t$, the data is encoded and stored in memory. Even though the encoded episodes require less memory than the real images, the system can still run out of memory when managing a continuous stream of incoming tasks. To cope with this issue, we propose a process inspired by *memory integration* in the hippocampus and the neocortex (Mack et al., 2018). Memory integration combines a new episode with a previously learned episode summarizing the information in both episodes in a single representation. The original episodes themselves are forgotten.

Consider a system that can store a total of $K$ encoded episodes based on its available memory. Assume that at increment $t - 1$, the system has a total of $K_{t-1}$ encoded episodes stored. It is now required to store $K_t$ more episodes in increment $t$. The system runs out of memory because $K_t + K_{t-1} > K$. Therefore, it must reduce the number of episodes to $K_r = K_{t-1} + K_t - K$. Because each task is composed of a set of classes at each increment, we reduce the total encoded episodes belonging to different classes based on their previous number of encoded episodes. Formally, the reduction in the number of encoded episodes $N_y$ for a class $y$ is calculated as (whole number):

$$N_y(new) = N_y(1 - \frac{K_r}{K_{t-1}}) \qquad (4)$$

To reduce the encoded episodes to $N_y(new)$ for class $y$, inspired by the memory integration process, we use an incremental clustering process that combines the closest encoded episodes to produce cen-

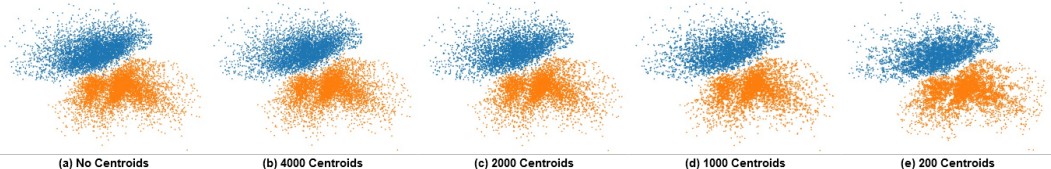

Figure 3: Visualization of the encoded episodes (a) and pseudo-encoded episodes generated using a different number of centroids and covariance matrices (b-e). Note that the pseudo-encoded episodes generated from a different number of centroids are all similar to the original encoded episodes. However, storing more centroids generates a feature space closer to the original feature space.

troids and covariance matrices. This clustering technique is similar to the *Agg-Var* clustering proposed in our earlier work (Ayub & Wagner, 2020b;a). The distance between encoded episodes is calculated using the Euclidean distance, and the centroids are calculated using the weighted mean of the encoded episodes. The process is repeated until the sum of the total number of centroids/covariance matrices and the encoded episodes for class $y$ equal $N_y(new)$. The original encoded episodes are removed and only the centroid and the covariances are kept (see Appendix A for more details).

### 3.3 REHEARSAL, PSEUDOREHEARSAL AND CLASSIFIER TRAINING

Figure 1 (b) depicts the procedure for training the classifier $D$ when new data belonging to task $t$ becomes available. The classifier is trained on data from three sources: 1) the real data for task $t$, 2) reconstructed images generated from the autoencoder's decoder, and 3) pseudo-images generated from centroids and covariance matrices for previous tasks. Source (2) uses the encodings from the previous tasks to generate a set of reconstructed images by passing them through the autoencoder's decoder. This process is referred to as rehearsal (Robins, 1995).

**Pseudorehearsal:** If the system also has old class data stored as centroids/covariance matrix pairs, pseudorehearsal is employed. For each centroid/covariance matrix pair of a class we sample a multivariate Gaussian distribution with mean as the centroid and the covariance matrix to generate a large set of pseudo-encoded episodes. The episodes are then passed through the autoencoder's decoder to generate pseudo-images for the previous classes. Many of the pseudo-images are noisy. To filter the noisy pseudo-images, we pass them through classifier $D$, which has already been trained on the prior classes, to get predicted labels for each pseudo-image. We only keep those pseudo-images that have the same predicted label as the label of the centroid they originated from. Among the filtered pseudo-images, we select the same number of pseudo-images as the total number of encoded episodes represented by the centroid and discard the rest (see Appendix A for the algorithm).

To illustrate the effect of memory integration and pseudorehearsal, we performed an experiment on MNIST dataset. We trained an autoencoder on 2 classes (11379 images) from MNIST dataset with the embedding dimension of size 2. After training, we passed all the training images for the two classes through the encoder to get 11379 encoded episodes (see Figure 3 (a)). Next, we applied our memory integration technique on the encoded episodes to reduce the encoded episodes to 4000, 2000, 1000 and 200 centroids (and covariances). No original encoded episodes were kept. Pseudorehearsal was then applied on the centroids to generate pseudo-encoded episodes. The pseudo-encoded episodes for different numbers of centroids are shown in Figure 3 (b-e). Note that the feature space for the pseudo-encoded episodes generated by different number of centroids is very similar to the original encoded episodes. For a larger number of centroids, the feature space looks almost exactly the same as the original feature space. For the smallest number of centroids (Figure 3 (e)), the feature space becomes less uniform and more divided into small dense regions. This is because a smaller number of centroids are being used to represent the overall concept of the feature space across different regions resulting in large gaps between the centroids. Hence, the pseudo-encoded episodes generated using the centroids are more dense around the centroids reducing uniformity. Still, the overall shape of the feature space is preserved after using the centroids and pseudorehearsal. Hence, our approach conserves information about previous classes, even with less memory, contributing to classifier performance while also avoiding catastrophic forgetting. The results presented in Section 4.3 on ImageNet-50 confirm the effectiveness of memory integration and pseudorehearsal.

**Sample Decay Weight:** The reconstructed images and pseudo-images can still be quite different from the original images, hurting classifier performance. We therefore weigh the loss term for reconstructed and pseudo-images while training $D$. For this, we estimate the degradation in the reconstructed and pseudo-images. To estimate degradation in the reconstructed images, we find the ratio of the classification accuracy of the reconstructed images ($c^r$) to the accuracy of the original images ($c^o$) on network $D$ trained on previous tasks. This ratio is used to control the weight of the loss term for the reconstructed images. For a previous task $t-1$, the weight $\Gamma_{t-1}^r$ (the sample decay weight) for the loss term $\mathcal{L}_{t-1}^r$ of the reconstructed images is defined as:

$$\Gamma_{t-1}^r = e^{-\gamma_{t-1}^r \alpha_{t-1}} \tag{5}$$

where $\gamma_{t-1}^r = 1 - \frac{c_{t-1}^r}{c_{t-1}^o}$ (*sample decay coefficient*) denotes the degradation in reconstructed images of task $t-1$ and $\alpha_{t-1}$ represents the number of times an autoencoder has been trained on the pseudo-images or reconstructed images of task $t-1$. The value of sample decay coefficient ranges from 0 to 1, depending on the classification accuracy of the reconstructed images $c_{t-1}^r$. If $c_{t-1}^r = c_{t-1}^o$, $\gamma_{t-1}^r = 0$ (no degradation) and $\Gamma_{t-1}^r = 1$. Similarly, the sample decay weight $\Gamma_{t-1}^p$ for loss term $\mathcal{L}_{t-1}^p$ for the pseudo-images is based on the classification accuracy $c_{t-1}^p$ of pseudo-images for task $t-1$. Thus, for a new increment, the total loss $\mathcal{L}_D$ for training $D$ on the reconstructed and pseudo-images of the old tasks and real images of the new task $t$ is defined as:

$$\mathcal{L}_D = \mathcal{L}_t + \sum_{i=1}^{t-1}(\Gamma_i^r \mathcal{L}_i^r + \Gamma_i^p \mathcal{L}_i^p) \tag{6}$$

## 4 EXPERIMENTS

We tested and compared EEC to several SOTA approaches on four benchmark datasets: MNIST, SVHN, CIFAR-10 and ImageNet-50. We also report the memory used by our approach and its performance in restricted memory conditions. Finally, we present an ablation study to evaluate the contribution of different components of EEC. Experiments on CIFAR-100 and comparison with additional methods are presented in Appendix D. Other generative memory approaches by previous authors did not test on the CIFAR-100 dataset.

### 4.1 DATASETS

The MNIST dataset consists of grey-scale images of handwritten digits between 0 to 9, with 50,000 training images, 10,000 validation images and 10,000 test images. SVHN consists of colored cropped images of street house numbers with different illuminations and viewpoints. It contains 73,257 training and 26,032 test images belonging to 10 classes. CIFAR-10 consists of 50,000 RGB training images and 10,000 test images belonging to 10 object classes. Each class contains 5000 training and 1000 test images. ImageNet-50 is a smaller subset of the iLSVRC-2012 dataset containing 50 classes with 1300 training images and 50 validation images per class. All of the dataset images were resized to 32×32, in concordance to (Ostapenko et al., 2019).

### 4.2 IMPLEMENTAION DETAILS

We used Pytorch (Paszke et al., 2019) and an Nvidia Titan RTX GPU for implementation and training of all neural network models. A 3-layer shallow convolutional autoencoder was used for all datasets (see Appendix B), which requires approximately 0.2MB of disk space for storage. For classification, on the MNIST and SVHN datasets the same classifier as the DCGAN discriminator (Radford et al., 2015) was used, for CIFAR-10 the ResNet architecture proposed by Gulrajani et al. (2017) was used and for ImageNet-50, ResNet-18 (He et al., 2016) was used.

In concordance with Ostapenko et al. (2019) (DGMw), we report the average incremental accuracy on 5 and 10 classes ($A_5$ and $A_{10}$) for the MNIST, SVHN and CIFAR-10 datasets trained continually with one class per increment. For ImageNet-50, the average incremental accuracy on 3 and 5 increments ($A_{30}$ and $A_{50}$) is reported with 10 classes in each increment. For a fair comparison,

|  | MNIST (%) | | SVHN (%) | | CIFAR-10 (%) | | ImageNet-50 (%) | |
|---|---|---|---|---|---|---|---|---|
| **Methods** | $A_5$ | $A_{10}$ | $A_5$ | $A_{10}$ | $A_5$ | $A_{10}$ | $A_{30}$ | $A_{50}$ |
| JT | 99.87 | 99.24 | 95.56 | 94.14 | 85.50 | 77.82 | 57.35 | 49.88 |
| iCaRL-S (Rebuffi et al., 2017) | 84.61 | 55.80 | - | - | 57.30 | 43.69 | 29.38 | 28.98 |
| RWalk-S (Chaudhry et al., 2018) | - | 82.50 | - | - | - | - | - | - |
| EEIL-S (Castro et al., 2018) | - | - | - | - | - | - | 27.87 | 11.80 |
| EWC-M (Seff et al., 2017) | 70.62 | 77.03 | 39.84 | 33.02 | - | - | - | - |
| DGR (Shin et al., 2017) | 90.39 | 85.40 | 61.29 | 47.28 | - | - | - | - |
| MeRGAN (Wu et al., 2018a) | 99.15 | 96.83 | 80.90 | 66.78 | - | - | - | - |
| DGMw (Ostapenko et al., 2019) | 98.75 | 96.46 | 83.93 | 74.38 | 64.94 | 56.21 | 32.14 | 17.82 |
| **EEC (Ours)** | **99.20** | **97.83** | **95.29** | **89.59** | **85.12** | **66.91** | **45.39** | **35.24** |
| Difference | **+0.05** | **+1.00** | **+11.3** | **+15.2** | **+20.2** | **+10.7** | **+13.2** | **+6.26** |
| EECS (Ours) | 98.00 | 96.26 | 94.30 | 81.12 | 82.20 | 61.91 | 41.13 | 30.89 |

Table 1: Comparison of EEC and EECS to approaches that store and use real samples (denoted with an S) and those that generate samples, for class-incremental learning on MNIST, SVHN, CIFAR-10 and ImageNet-50 datasets. The difference between EEC and the SOTA approach is shown in bold.

we typically compare against approaches with a generative memory replay component that do not use a pre-trained feature extractor and are evaluated in a single-headed fashion. Among such approaches, to the best of our knowledge, DGMw represents the state-of-the-art benchmark on these datasets which is followed by MeRGAN (Wu et al., 2018a), DGR (Shin et al., 2017) and EWC-M (Seff et al., 2017). Joint training (JT) is used to produce an upperbound for all four datasets. We compare these methods against two variants of our approach: EEC and EECS. EEC uses a separate autoencoder for classes in a new increment, while EECS uses a single autoencoder that is retrained on the reconstructed images of the old classes when learning new classes. For both EEC and EECS, results are reported when all of the encoded episodes for the previous classes are stored. The results for EEC under restricted memory conditions are also presented.

Hyperparameter values and training details are reported in Appendix C. We performed each experiment 10 times with different random seeds and report average accuracy over all runs.

## 4.3 COMPARISON WITH SOTA METHODS

Table 1 compares EEC and EECS against SOTA approaches on the MNIST, SVHN, CIFAR-10 and ImageNet-50 datasets. We compare against two different types of approaches, those that use real images (episodic memory) of the old classes and those that generate previous class images when learning new classes. Both EEC and EECS outperform EWC-M and DGR by significant margins on the MNIST on $A_5$ and $A_{10}$. MeRGAN and DGMw perform similarly to our methods on the $A_5$ and $A_{10}$ experiments. Note that MeRGAN and EEC approach the JT upperbound on $A_5$. Further, accuracy for MeRGAN, DGMw and EEC changes only slightly between $A_5$ and $A_{10}$, suggesting that MNIST is too simple of a dataset for testing continual learning using generative replay.

Considering SVHN, a more complex dataset consisting of colored images of street house numbers, the performance of our method remains reasonably close to the JT upperbound, even though the performance of other approaches decreases significantly. For $A_5$, EEC achieves only **0.27%** lower accuracy than JT and **11.36%** higher than DGMw (current best method). For $A_{10}$, EEC is 4.55% lower than JT but still **15.21%** higher than DGMw and achieves **5.66%** higher accuracy on $A_{10}$ than DGMw did on $A_5$. EECS performs slightly lower than EEC on $A_5$ but the gap widens on $A_{10}$. However, EECS also beats the SOTA approaches on both $A_5$ and $A_{10}$.

Considering the more complex CIFAR-10 and ImageNet-50 datasets, only DGMw reported results for these datasets. On CIFAR-10, EEC beats DGMw on $A_5$ by a margin of **20.18%** and on $A_{10}$ by a margin of **10.7%**. Similar to the SVHN results, the accuracy achieved by EEC on $A_{10}$ is even higher than DGMw's accuracy on $A_5$. In comparison with the JT, EEC performs similarly on $A_5$ (**0.38%** lower), however on $A_{10}$ it performs significantly lower. For ImageNet-50, again we see similar results as both EEC and EECS outperform DGMw on $A_{30}$ and $A_{50}$ by significant margins (**13.29%** and **17.42%**, respectively). Similar to SVHN and CIFAR-10 results, accuracy for EEC on $A_{50}$ is even higher than DGMw's accuracy on $A_{30}$. Further, EEC also beats iCaRL (episodic

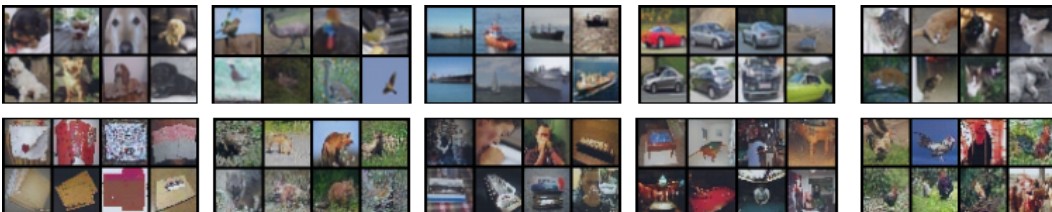

Figure 4: Reconstructed Images for CIFAR-10 (top) and ImageNet-50 (bottom) after all tasks

memory SOTA method) with margins of **16.01%** and **6.26%** on $A_{30}$ and $A_{50}$, respectively, even though iCaRL has an unfair advantage of using stored real images.

**Discussion:** Our method performed consistently across all four datasets, especially on $A_5$ for MNIST, SVHN and CIFAR-10. In contrast, DGMw (the best current method) shows significantly different results across the four datasets. The results suggest that the current generative memory-based SOTA approaches are unable to mitigate catastrophic forgetting on more complex RGB datasets. This could be because GANs tend to generate images that do not belong to any of learned classes, which can drastically reduce classifier performance. Our approach copes with these issues by training autoencoders borrowing ideas from the NST algorithm and retraining of the classifier with sample decay weights. Images reconstructed by EEC for CIFAR-10 after 10 tasks and for ImageNet-50 after 5 tasks are shown in Figure 4 (more examples in supplementary file).

**Memory Usage Analysis:** Similar to DGMw, we analyze the disc space required by our model for the ImageNet-50 dataset. For EEC, the autoencoders use a total disc space of 1 MB, ResNet-18 uses about 44 MB, while the encoded episodes use a total of about 66.56 MB. Hence, the total disc space required by EEC is about 111.56 MB. DGMw's generator (with corresponding weight masks), however, uses 228MB of disc space and storing pre-processed real images of ImageNet-50 requires disc space of 315MB. Hence, our model requires **51.07%** ((228-111.56)/228 = 0.5107) less space than DGMw and **64.58%** less space than the real images for ImageNet-50 dataset.

EEC was tested with different memory budgets on the ImageNet-50 dataset to evaluate the impact of our memory management technique. The memory budgets (K) are defined as the sum of the total number of encoded episodes, centroids and covariance matrices (diagonal entries) stored by the system. Figure 5 shows the results in terms of $A_{30}$ and $A_{50}$ for a wide range of memory budgets. The accuracy of EEC changes only slightly over different memory budgets. Even with a low budget of K=5000, the $A_{30}$ and $A_{50}$ accuracies are only **3.1%** and **3.73%** lower than the accuracy of EEC with unlimited memory. Further, even for K=5000, EEC beats DGMw (current SOTA on ImageNet-50) by margins of **10.17%** and **13.73%** on $A_{30}$ and $A_{50}$, respectively. The total disc space required for K=5000 is only 5.12 MB and the total disc space for the complete system is 50.12 MB (44 MB for ResNet-18 and 1 MB for autoencoders), which is **78.01%** less than DGMw's required disc space (228 MB). These results clearly depict that our approach produces the best results even with extremely limited memory, a trait that is not shared by other SOTA approaches. Moreover, the results also show that our approach is capable of dealing with the two main challenges of continual learning mentioned earlier: catastrophic forgetting and memory management.

### 4.4 ABLATION STUDY

We performed an ablation study to examine the contribution of using: 1) the content loss while training the autoencoders, 2) pseudo-rehearsal and 3) the sample weight decay. These experiments were performed on the ImageNet-50 dataset. Complete encoded episodes of previous tasks were used for ablations 1 and 3, and a memory budget of K=5000 was used for ablation 2. We created hybrid versions of EEC to test the contribution of each component. Hybrid 1: termed as EEC-noNST does not use the content loss while training the autoencoders. Hybrid 2: termed as EEC-CGAN uses a conditional GAN instead of NST based autoencoder. Hybrid 3: termed as EEC-VAE uses a variational autoencoder and Hybrid 4: termed as EEC-DNA uses a denoising autoencoder instead of our proposed NST based autonecoder. Hybrid 5: termed as EEC-noPseudo simply removes the extra encoded episodes when the system runs out of memory and does not use pseudo-rehearsal. Hybrid 6: termed as EEC-noDecay does not use sample weight decay during classifier training on new and

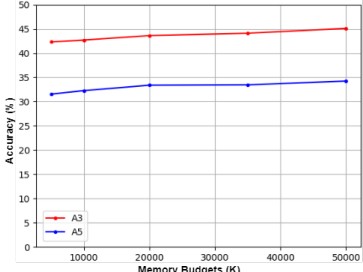

| Methods | $A_{30}$ | $A_{50}$ |
|---|---|---|
| EEC-noNST | 39.51 | 28.73 |
| EEC-CGAN | 39.94 | 29.12 |
| EEC-VAE | 39.14 | 27.63 |
| EEC-DNA | 39.42 | 28.36 |
| EEC-noDecay | 41.62 | 30.47 |
| **EEC** | **45.39** | **35.24** |
| EEC-noPseudo | 37.31 | 25.24 |
| **EEC (K=5000)** | **42.29** | **31.51** |

Figure 5: The accuracy of EEC on $A_{30}$ and $A_{50}$ on ImageNet-50 dataset with different memory budgets.

Table 2: Ablation study results in terms of $A_{30}$ and $A_{50}$ for ImageNet-50 dataset.

old tasks. Except for the changed component, all the other components in the hybrid approaches were the same as used in EEC.

All of the hybrids show inferior performance as compared to the complete EEC approach (Table 2). Evaluated on $A_{30}$ and $A_{50}$, EEC-noNST, EEC-CGAN, EEC-VAE and EEC-DNA all result in ~5.8% and ~6.5% lower accuracy, respectively than EEC. EEC-noDecay results in 3.77% and 4.77% lower accuracy on $A_{30}$ and $A_{50}$ respectively, than EEC. As a fair comparison with EEC using K=5000, EEC-noPseudo results in 4.98% and 6.27% lower accuracy for $A_{30}$ and $A_{50}$, respectively. The results show that all of these components contribute significantly to the overall performance of EEC, however training the autoencoders with content loss and pseudo-rehearsal seem to be more important. Also, note that the conditional GAN with the other components of our approach (such as sample weight decay) produces **7.80%** and **11.3%** higher accuracy on $A_{30}$ and $A_{50}$, respectively, than DGMw. Further note that all the hybrids that do not use the NST based autoencoder (Hybrids 1-4) produce similar accuracy, since they use the sample weight decay to help mange the degradation of reconstructed images. These results show the effectiveness of sample weight decay to cope with image degradation.

### 4.5 EXPERIMENTS ON HIGHER RESOLUTION IMAGES

As mentioned in Subsection 4.2, all the experiments on the four datasets were done using resized images of size 32×32. To show further insight into our approach, we performed experiments on ImageNet-50 dataset with higher resolution images of size 256×256 and randomly cropped to 224×224. Evaluated on $A_{30}$ and $A_{50}$, EEC achieved **70.75%** and **56.89%** accuracy, respectively using images of size 224×224. These results show that using higher resolution images improve the performance of EEC by significant margins (25.36% and 21.65% increase on $A_{30}$ and $A_{50}$). Note that storing real high resolution images requires a much larger disk space than storing images of size 32×32. For ImageNet-50, storing original images of size 224×224 requires 39.13GB, while images of size 32×32 require only 315MB. In contrast, EEC requires only 13.04 GB (66% less than real images of size 224×224) when using the same autoencoder architecture that was used for images of size 32×32. Using a different architecture, we can bring the size of encoded episodes to be the same as for images of size 32×32 but the accuracy achieved is lower. These results further show the effectiveness of our approach to mitigate catastrophic forgetting while using significantly less memory compared to real images.

## 5 CONCLUSION

This paper has presented a novel and potentially powerful approach (EEC) to strict class-incremental learning. Our paper demonstrates that the generation of high quality reconstructed data can serve as the basis for improved classification during continual learning. We further demonstrate techniques for dealing with image degradation during classifier training on new tasks as well as a cognitively-inspired clustering approach that can be used to manage the memory. Our experimental results demonstrate that these techniques mitigate the effects of catastrophic forgetting, especially on complex RGB datasets, while also using less memory than SOTA approaches. Future continual learning approaches can incorporate different components of our approach such as the NST-based autoencoder, pseudo-rehearsal and sample decay weights for improved performance.

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

## A   EEC ALGORITHMS

The algorithms below describe portions of the complete EEC algorithm. Algorithm 1 is for autoencoder training (Section 3.1 in paper), Algorithm 2 is for memory integration (Section 3.2 in paper), Algorithm 3 is for rehearsal, pseudo-rehearsal and classifier training (Section 3.3 in paper) and Algorithm 4 is for filtering pseudo-images (Section 3.3 in paper).

---

### Algorithm 1: EEC: Train Autoencoder

---

**Input:** $S_t = \{(x_i^t, y_i^t)\}_{i=1}^{n^t}$       ▷ data points with ground truths for task $t$
**require:** $\lambda$
**require:** $f_r = (g_{dec} \circ g_{enc})$       ▷ autoencoder pipeline
**require:** $f_c : \mathcal{X} \to \mathcal{F}_c$       ▷ convolutional feature extractor from classifier $D$
**require:** $N_{epoch}$       ▷ Number of epochs
**Output:** $F_t = \{(f_i^t, y_i^t)\}_{i=1}^{n^t}$       ▷ encoded episodes with ground truths for task $t$

1: **for** $j = 1; j < N_{epoch}$ **do**
2:      $X_t^* = f_r(X^t)$       ▷ get reconstructed images for task $t$
3:      $\mathcal{L}_r = ||X_t - X_t^*||_2$       ▷ Reconstruction loss
4:      $\mathcal{L}_{cont} = ||f_c(X_t) - f_c(X_t^*)||_2$
5:                                         ▷ Content loss
6:      $\mathcal{L} = (1 - \lambda)\mathcal{L}_r + \lambda\mathcal{L}_{cont}$       ▷ complete autoencoder loss
7:      $\theta_r \leftarrow minimize(\mathcal{L})$       ▷ update autoencoder parameters such that it minimizes $\mathcal{L}$
8: $F_t = g_{enc}(X_t)$       ▷ get encoded episodes for images of task $t$

---

### Algorithm 2: EEC: Memory Integration

---

**Input:** $F = \{F_i\}_{i=1}^{t-1}$, where $F_i = \{(f_j^i, y_j^i)\}_{j=1}^{n^i}$       ▷ encoded episodes set for previous $t - 1$ tasks
**require:** $K$       ▷ maximum number of episodes
**require:** $K_t$       ▷ number of encoded episodes for task $t$
**Output:** $F(new)$       ▷ reduced encoded episodes for previous tasks
**Output:** $C = \{C_i\}_{i=1}^{t-1}$       ▷ centroids for previous tasks
**Output:** $\Sigma = \{\Sigma_i\}_{i=1}^{t-1}$       ▷ covariances for previous tasks
**Initialize:** $K_{t-1} = \sum_{i=1}^{t-1} n^i$       ▷ number of encoded episodes for $t - 1$ previous tasks

1: $K_r = K_t + K_{t-1} - K$
2: **for** $i = 1; i \leq t - 1$ **do**
3:      **for** $j = 1; j \leq y^t$ **do**
4:          **repeat**       ▷ for each class in task $i$ find centroids and covariances
5:             $c_j, \sigma_j \leftarrow combine(x_l^i, x_q^i) \forall y_l^i = y_q^i = j$ ▷ Combine closest points for class $j$ in task $i$
6:             $N_j(new) = 2N_j(cent) + \sum_{l=1, y_l^i = j}^{n^i} 1$ ▷ $N_j(cent)$: number of centroids for class $j$
7:          **until** $N_j(new) = N_j(1 - \frac{K_r}{K_{t-1}})$

---

---

Algorithm 3: EEC: Train Classifier

---

**Input:** $S_t = \{(x_i^t, y_i^t)\}_{i=1}^{n^t}$        ▷ data points with ground truths for task $t$

**Input:** $F = \{F_i\}_{i=1}^{t-1}$, where $F_i = \{(f_j^i, y_j^i)\}_{j=1}^{n^i}$     ▷ encoded episodes for previous tasks

**Input:** $C = \{C_i\}_{i=1}^{t-1}$, where $C_i = \{(c_j^i, y_j^i)\}_{j=1}^{N^i(cent)}$     ▷ centroids for previous tasks

**Input:** $\Sigma = \{\Sigma_i\}_{i=1}^{t-1}$, where $\Sigma_i = \{(\sigma_j^i, y_j^i)\}_{j=1}^{N^i(cent)}$    ▷ covariances for previous tasks

**Input:** $M = \{M_i\}_{i=1}^{t-1}$, where $M_i = \{m_j^i\}_{j=1}^{N^i(cent)}$    ▷ no. of episodes clustered in each centroid

**require:** $c_{t-1}^o$        ▷ accuracy of original training images of previous $t-1$ tasks

**require:** $D$        ▷ Classifier model

**require:** $\{g_{dec}^j\}_{j=1}^{t-1}$        ▷ decoder part of autoencoders for each of the previous tasks

**require:** $N_{epoch}$        ▷ Number of epochs

1: **for** $i = 1; i \leq t - 1$ **do**
2:      **for** $j = 1; j \leq N^i(cent)$ **do**
3:          $F_i^*(large).append(multivariate\_normal(c_j^i, \sigma_j^i))$    ▷ generate pseudo-samples for each concept
4:      $X_i^{'}(large) = g_{dec}^i(F_i^*(large))$ ▷ generate large no. of pseudo-images from pseudo-samples
5:      $X_i^{'} = FILTER(X_i^{'}(large), M_i)$        ▷ Algorithm 4
6:      $X_i^* = g_{dec}^i(F_i)$        ▷ get reconstructed images for encoded episodes
7: **for** $i = 1; i \leq t - 1$ **do**
8:      $Y_i^* = D(X_i^*)$        ▷ predictions for reconstructed images for task $i$
9:      $Y_i^{'} = D(X_i^{'})$        ▷ predictions for pseudo-images for task $i$
10:      $c_i^r = \frac{\sum_{j=1}^{n^{i*}} 1[Y_i^*(j)==Y_i(j)]}{n^{i*}}$        ▷ accuracy for reconstructed images of task $i$
11:      $c_i^p = \frac{\sum_{j=1}^{n^{i'}} 1[Y_i^{'}(j)==Y_i(j)]}{n^{i'}}$        ▷ accuracy for pseudo images of task $i$
12:      $\gamma_i^r = 1 - \frac{c_i^r}{c_i^o}$        ▷ sample decay coefficient for reconstructed images of task $i$
13:      $\gamma_i^p = 1 - \frac{c_i^p}{c_i^o}$        ▷ sample decay coefficient for pseudo images of task $i$
14:      $\Gamma_i^r = e^{-\gamma_i^r \alpha_i}$
15:      $\Gamma_i^p = e^{-\gamma_i^p \alpha_i}$
16: **for** $j = 1; j < N_{epoch}$ **do**
17:      **for** $i = 1; i \leq t - 1$ **do**
18:          $Y_i^* = D(X_i^*)$        ▷ predictions for reconstructed images for task $i$
19:          $Y_i^{'} = D(X_i^{'})$        ▷ predictions for pseudo-images for task $i$
20:          $\mathcal{L}_i^r = CrossEntropy(Y_i^* - Y_i)$        ▷ loss for reconstructed images of task $i$
21:          $\mathcal{L}_i^p = CrossEntropy(Y_i^{'} - Y_i)$        ▷ loss for psuedo-images of task $i$
22:      $Y_t^* = D(X_t)$        ▷ predictions for data points for task $t$
23:      $\mathcal{L}_t = CrossEntropy(Y_t^* - Y_t)$        ▷ loss for task $t$
24:      $\mathcal{L}_D = \mathcal{L}_t + \sum_{i=1}^{t-1} \Gamma_i^r \mathcal{L}_i^r + \Gamma_i^p \mathcal{L}_i^p$
25:      $\theta_D \leftarrow minimize(\mathcal{L}_D)$        ▷ update classifier parameters such that it minimizes $\mathcal{L}_D$

---

Algorithm 4: EEC: Filter Pseudo-images

---

**Input:** $S_t(large) = \{(x_i^t, y_i^t)\}_{i=1}^{n^t(large)}$, where $X_t(large) = \{x_i^t\}_{i=1}^{n^t(large)}$,
$Y_t(large) = \{y_i^t\}_{i=1}^{n^t(large)}$

**Input:** $M_t = \{m_i^t\}_{i=1}^{N^t(cent)}$

**require:** $D$

**Output:** $S_t = \{(x_i^t, y_i^t)\}_{i=1}^{n^t}$        ▷ filtered pseudo-images for task $t$

1: $Y_t^*(large) = D(X_t(large))$
2: $X_t = \{x_i^t \in X_t(large); y_i^t == y_i^{t*}\}_{i=1}^{n^t(large)}$        ▷ pseudo-images with correct predicted label
3: $X_t = \{x_i^t \in X_t\}_{i=1}^{M_t}$        ▷ keep $M_t$ pseudo-images

---

## B AUTOENCODER ARCHITECTURES

|  | Name | Layer Type | Filter Size | Stride | Output Shape |
|---|---|---|---|---|---|
| | conv1 | Conv2D | 3×3 | 2 | (None,64,16,16) |
| | batch_norm1 | BatchNorm2d | - | - | (None,64,16,16) |
| | relu1 | ReLU | - | - | (None,64,16,16) |
| | conv2 | Conv2D | 3×3 | 2 | (None,32,8,8) |
| | batch_norm2 | BatchNorm2d | - | - | (None,32,8,8) |
| Encoder | relu2 | ReLU | - | - | (None,32,8,8) |
| | conv3 | Conv2D | 3×3 | 2 | (None,16,4,4) |
| | batch_norm3 | BatchNorm2d | - | - | (None,16,4,4) |
| | relu3 | ReLU | - | - | (None,16,4,4) |
| | conv_trans3 | ConvTranspose2D | 3×3 | 2 | (None,32,8,8) |
| | batch_norm3 | BatchNorm2d | - | - | (None,32,8,8) |
| | relu3 | ReLU | - | - | (None,32,8,8) |
| | conv_trans2 | ConvTranspose2D | 3×3 | 2 | (None,64,16,16) |
| | batch_norm2_ | BatchNorm2d | - | - | (None,64,16,16) |
| Decoder | relu2_ | ReLU | - | - | (None,64,16,16) |
| | conv_trans1 | ConvTranspose2D | 3×3 | 2 | (None,1,32,32) |
| | batch_norm1_ | BatchNorm2d | - | - | (None,1,32,32) |
| | relu1_ | ReLU | - | - | (None,1,32,32) |

Table 3: Convolutional Autoencoder Architecture for MNIST, SVHN, CIFAR-10 and ImageNet-50 datasets

## C HYPERPARAMETERS

| Hyperparameters | Values |
|---|---|
| No. of epochs | 100 |
| Starting Learning Rate (LR) | 0.001 |
| LR Decay Milestones | {50} |
| LR Decay Factor | 1/10 |
| miniBatch Size | 128 |
| Optimizer | Adam |
| Weight Decay | 0.0005 |
| $\lambda$ | 0.7 |

Table 4: Hyper-parameters for EEC autoencoder training

| Hyperparameters | Values |
|---|---|
| No. of epochs (1st increment) | 200 |
| No. of epochs (next increments) | 45 |
| Starting Learning Rate (LR) | 0.1 |
| LR Decay Milestones | {60,120,160} |
| LR Decay Factor | 1/5 |
| miniBatch Size | 128 |
| Optimizer | SGD |
| Momentum | 0.9 |
| Weight Decay | 0.0005 |

Table 5: Hyper-parameters for EEC classifier training

| Methods | Accuracy (%) |
|---|---|
| iCaRL (Rebuffi et al., 2017) | 59.5 |
| EEIL (Castro et al., 2018) | 64.0 |
| BiC (Wu et al., 2019) | 64.3 |
| LUCIR (Hou et al., 2019) | 63.4 |
| **EEC (Ours)** | **65.6** |
| FearNet (Kemker & Kanan, 2018) | 66.2 |
| CAN (Xiang et al., 2019) | 67.1 |
| **EEC-P (Ours)** | **76.7** |

Table 6: Comparison of EEC against episodic memory approaches and approaches that use a pre-trained CNN in terms of average incremental accuracy (%) with 10 classes per increment. EEC-P uses a pre-trained CNN for feature extraction.

## D  EXPERIMENTS ON CIFAR-100

For a fair comparison, the main set of approaches that we compared to are generative memory approaches. These approaches were only tested on the four datasets (Section 4). However, many episodic memory approaches and approaches that use pre-trained CNN features did not present results on the four datasets in Section 4. Hence, we test our approach (EEC) on CIFAR-100 to compare against more episodic memory approaches and approaches that use a pre-trained CNN as a feature extractor.

CIFAR-100 consists of 60,000 $32 \times 32$ images belonging to 100 object classes. There are 500 training images and 100 test images for each class. We divided the dataset into 10 batches with 10 classes per batch. During training, in each increment we provided the algorithm a new batch of 10 classes. We report the average incremental accuracy over the 10 increments as the evaluation metric. For evaluation against approaches that use a pre-trained network, we first extracted features from a pre-trained ResNet-34 on ImageNet and then applied EEC on the feature vectors instead of raw RGB images. This version of EEC is denoted as EEC-P. The hyperparmeter values and training details for this experiment are presented in Appendix C. We performed this experiment 10 times with different random seeds and report the average accuracy over all the runs.

Among the episodic memory approaches, we compare against iCaRL (Rebuffi et al., 2017), EEIL (Castro et al., 2018), BiC (Wu et al., 2019) and LUCIR (Hou et al., 2019). Among the approaches that use a pre-trained CNN, we compare against FearNet (Kemker & Kanan, 2018) and the method proposed by Xiang et al. (2019) (CAN). These approaches have been introduced in Section 2.

Table 6 shows the comparison of EEC against all the above mentioned approaches. EEC beats all the state-of-the-art (SOTA) episodic memory approaches by a significant margin, even though these approaches have an unfair advantage of using stored real images of the old classes. Although the difference between EEC and other approaches (1.3%) is lower than the difference on the other four datasets. EEC-P also beats the SOTA approaches that use a pre-trained CNN by a much greater margin (**9.6%**). These results further show the effectiveness of our approach for mitigating catastrophic forgetting.

## E  COMPARISON WITH FEARNET

FearNet (Kemker & Kanan, 2018) is one of the few approaches that uses some ideas similar to EEC. In particular, FearNet uses autoencoders to reconstruct old data and stores a single centroid and covariance matrix for each of the old classes. FearNet also uses pseudorehearsal on the centroids and covariances of old classes to regenerate pseudo-encoded episodes of the old classes, which is the same as our approach. However, unlike EEC, FearNet uses a pre-trained feature extractor which is a limitation. Further, for pseudorehearsal, FearNet only stores a single centroid and covariance matrix for each of the old classes, which is not enough to capture the entire feature space of the classes. In contrast, EEC uses memory integration based clustering technique to store multiple centroids and covariance matrices per class to better capture the overall feature space of the old classes.

To illustrate the difference between the two approaches, we performed the same experiment on MNIST dataset as in Section 3.3. For FearNet, we allowed the model to store a single centroid (and

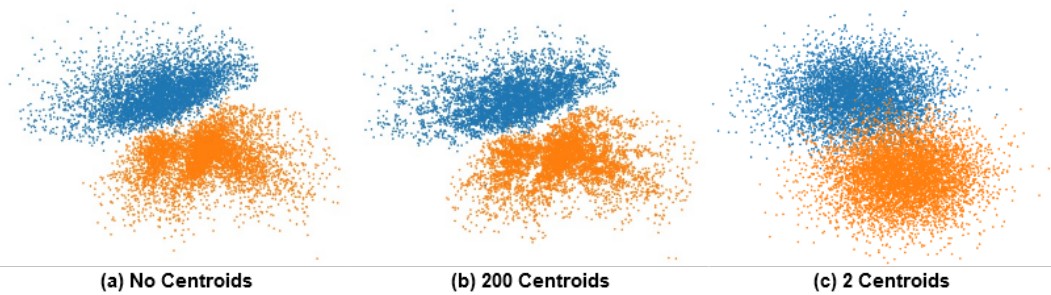

Figure 6: Visualization of the encoded episodes (a), pseudo-encoded episodes generated using 200 centroids and covariance matrices (b) and pseudo-encoded episodes generated using 2 centroids and covariance matrices (c).

| Methods | $A_{30}$ | $A_{50}$ | Memory (MB) |
|---|---|---|---|
| JT | 57.38 | 49.88 | 315 |
| EEC ($r = 0.25$) | 54.85 | 45.92 | 161.48 |
| EEC ($r = 0.2$) | 54.04 | 45.14 | 151.496 |
| EEC ($r = 0.15$) | 53.38 | 44.26 | 141.412 |
| EEC ($r = 0.1$) | 51.67 | 42.94 | 131.53 |
| EEC ($r = 0$) | 45.39 | 35.24 | 111.56 |

Table 7: Performance of EEC for different values of $r$

covariance matrix) for each of the two classes and then used pseudorehearsal to generate pseudo-encoded episodes. Figure 6 shows the comparison between the original feature space (Figure 6 (a)), pseudo-encoded episodes generated with 200 centroids using memory integration (Figure 6 (b)) and pseudo-encoded episodes generated using a single centroid per class (2 centroids) as in FearNet (Figure 6 (c)). The feature space generated by FearNet is almost circular and does not resemble the original feature space. Also, there is an overlap between the feature spaces of the two classes which will lead to similar images for the two classes which will eventually hurt classifier performance. In contrast, the pseudo-encoded episodes generated by memory integration capture the overall concept of the original feature space without any overlap between the feature spaces of the two classes. Results in Table 6 on CIFAR-100 dataset also confirm that EEC is significantly superior to FearNet in terms of the average incremental accuracy (**10.5%** improvement in accuracy).

## F    EXPERIMENT ON IMAGENET-50 WITH ORIGINAL AND RECONSTRUCTED IMAGES

In this experiment, we allow EEC to partially store original images and the rest as encoded episodes. We define a ratio of original images and encoded episodes as $r = n_o/N$, where $n_o$ is the number of original images stored per class and $N$ is the total number of images (original and reconstructed) per class. This experiment was performed with images of size 32×32. Table 7 shows the accuracy for EEC in terms of $A_{30}$ and $A_{50}$ for different values of $r$. $r = 0$ shows EEC with no original images and only the encoded episodes. By increasing $r$ to 0.1, we see a dramatic increase in accuracy, however the increase in memory is minimal (131.53 MB for $r = 0.1$ as compared to 111.56 MB for $r = 0$). As the value of $r$ continues to increase, EEC's accuracy gets closer and closer to the JT upperbound.

