# OpenReview forum: "EEC: Learning to Encode and Regenerate Images for Continual Learning"
_ICLR.cc/2021/Conference — ICLR 2021 Poster_

### Official Review · AnonReviewer2 · 2020-10-23
**The idea of paper is quite similar to existing method FearNet using autoencoders. The experiments are limited to small images and  a few tasks.**

**Rating:** 4
**Confidence:** 5

**Review:**

Summary:

The paper tackles catastrophic forgetting for continual learning. It proposes to train autoencoders with Neural Style Transfer to generate previous images. The encoded episodics can be converted into centroids and covariances matrices in order to save memory usage. It shows significant improvements in the experimental parts.

Strengths:

- The results on tiny datasets (MNIST, SVHN and CIFAR-10) are promising. On Tiny ImageNet-50, the performance compared to other methods looks good, but it is relatively bad compared to some other datasets.
- It is well written and easy to follow.


Concerns:

- The overall idea is very close to what FearNet proposed by using autoencoders. The main difference is to use autoencoders on images instead of features.  Neural Style Transfer is helpful to generate more realistic images, therefore it makes sense to help during incremental learning to replay previous knowledge. However, it seems obvious improving image quality will result in better performance for methods based on replay.

- From the experimental results compared to other methods, it is still limited to relative small resolution images and a few tasks. It would be interesting to see the performance on more challenging datasets with large resolution and more tasks.

---

> ### Author Response · Authors · 2020-11-14
> **Concerns addressed - Experiment on high resolution images added**
>
> We thank you for your helpful comments, in particular to test our approach on datasets with high resolution. The new experiment has shown further insights into the effectiveness of our approach for continual learning.
>
> Concerns:
>
> 1) it seems obvious improving image quality will result in better performance for methods based on replay:
> We agree that improving image quality is an obvious way to improve performance. However, improving image quality is not a simple task, especially when using limited memory. As shown in the results from earlier papers (Shin et al. 2017, Wu et al. 2018a, Ostapenko et al. 2019), the regenerated images are not even close to the original images, hence the overall accuracy decreases drastically, especially for ImageNet-50 even with images of size 32$\times$32. Thus, in this paper, we propose the NST based autoencoder to improve image quality while utilizing lower memory. Further, note that the reconstructed images cannot be the same as original images, especially when using limited memory. Thus, we introduce weight decay and image filtering techniques (Section 3.3) to deal with image degradation during reconstruction so that degraded images do not hurt classifier performance.
> FearNet does not use any of these ideas. FearNet simply uses a network pre-trained on ImageNet to extract features, which is a limitation since it can only be applied on object-centric image datasets. Further, using the pre-trained feature extractor gives FearNet an unfair advantage over other approaches. The only similarity between EEC and FearNet, other than the use of autoencoders, is that they both use pseudorehearsal. However, FearNet only uses a single centroid and covariance matrix to represent the feature space of a class. In contrast we use memory integration-based clustering approach to better cover the complete feature space. We have added a new experiment in Appendix E to compare pseudorehearsal in EEC and FearNet. This experiment demonstrates that unlike FearNet, EEC captures the overall concept of the original feature space. Experiments on CIFAR-100 (Appendix D) show that EEC outperforms FearNet by a margin of 10.5\%, which shows the effectiveness of our approach in comparison with FearNet.
> 2)  It would be interesting to see the performance on more challenging datasets with large resolution and more tasks:
> The only reason to use the four datasets (with images of size 32$\times$32) in the paper was to have a fair comparison with other generative memory based approaches. However, to have a comparison with more episodic memory based approaches on a larger dataset with more classes, we added an experiment on CIFAR-100 in Appendix D. These results show that EEC outperforms other SOTA approaches even on a larger dataset.
> To test EEC on images with higher resolution, we have added a new experiment in the paper (Section 4.5). We performed the experiment on ImageNet-50 dataset with images of size 224$\times$224. The results show that EEC produces significantly higher accuracy when using higher resolution images than when using lower resolution images. Further, the memory required for EEC remains the same regardless of the image resolution, which demonstrates the effectiveness of EEC to mitigate catastrophic forgetting while using significantly less memory that other approaches.

---

### Official Review · AnonReviewer3 · 2020-10-27
**Good Paper**

**Rating:** 6
**Confidence:** 2

**Review:**

##########################################################################

Summary:


The paper proposes an approach which trains autoencoders with Neural Style Transfer to encode and store images. The method is applied to the problem of continual learning to overcome the catastrophic forgetting and memory limitation on the storage data. The authors report that the presented approach increases the classification accuracy by 13-17% over SOTA methods

##########################################################################

Reasons for score:


Overall, I vote for accepting. The paper presents nice ideas, outperforms SOTA methods and is clearly written. The contribution could have been stronger with a more detailed evaluation and better presentation.

##########################################################################

Pros:


1. I really like the idea of using NST to train the autoencoder.

2. Clarity of the paper.


##########################################################################

Cons:

1. The system uses a pretraied Style Transfer Network with Imagenet. Does it offer an unfair advantage over other approaches?
2. It would be great to compare the NST Autoencoder with Variational or Denoising Autoencoder.
3. The validation has been performed using low-resolution images (32x32)
4. The improvement on Imagnet-50 A5 against iCaRL-S is 6.28. So, the difference should be 6.28 and not 17.4.
5. It would be great to see the results for A30 and A50 with imagenet as it was done by (Ostapenko et.al. 2019)
6. Figure 4 shows the accuracy on ImageNet-50 with different budgets. It seems that the accuracy is still increasing, did you tried with larger values?

Minor points:
- Table 1 - it would be great to include the reference paper of each method



##########################################################################

Questions during rebuttal period:


Please address and clarify the cons above


#########################################################################

Some typos:

(1) Page 5: corariance -> covariance
(2) Page 7: the the accuracy --> the accuracy


Update after rebuttal:
My initial concerns were clarified by the authors and the paper substantially improved.

---

> ### Author Response · Authors · 2020-11-14
> **Cons and typos addressed in the paper - More experiments added with variational and denoising autoencoder**
>
> We thank you for your insightful comments and have used these comments to improve the paper.
>
> Cons:
> 1) We believe there is a misunderstanding regarding our approach. Our system does not use a network pre-trained on ImageNet as Style Transfer Network. The classifier network in our approach is first trained on the original images of the current increment and reconstructed images of the old classes (after the first increment). This network is then used as a fixed feature extractor during autoencoder training using the content loss.
> 2) Thank you for your suggestion. We have added the comparison with variational and denoising autoencoders in the ablation study (Section 4.4). The results demonstrate that our NST based autoencoder significantly outperforms the variational and denoising autoencoders.
> 3) The only reason to perform our experiments with low-resolution images (32$\times$32) was to have a fair comparison with previous approaches (Wu et al. 2018a, Ostapenko et al. 2019). We have added a new experiment in the paper (Section 4.5) on ImageNet-50 dataset with higher resolution images (224$\times$224), which further shows the effectiveness of EEC in continual learning.
> 4) We have updated the difference value in the paper to 6.26.
> 5) The results shown in the paper are for $A_{30}$ and $A_{50}$. We showed them as $A_3$ and $A_5$ to show 3 and 5 tasks instead of 30 and 50 classes learned. We have fixed this in the paper.
> 6) The largest value for the memory budget can be 65000 encoded episodes because ImageNet-50 contains 65000 images. The value reported in Table 1 for EEC uses the maximum budget. However, another way to increase the memory budget is to allow the model to store a small set of original images and the rest as encoded episodes. We have reported an experiment on ImageNet-50 dataset with different ratios of original images (Appendix F), which shows that the accuracy for EEC does continue to increase when using higher memory budgets.
>
> Minor Points:
> The references for the papers are added in Table 1.
>
> Typos:
> Fixed in the paper.

---

### Official Review · AnonReviewer1 · 2020-11-03
**Problem is not well motivated - some basic concerns - model is not principled enough**

**Rating:** 4
**Confidence:** 5

**Review:**

In continual learning settings, one of the important technique for avoiding catastrophe forgetting is to replay data points from the past. For memory efficiency purposes, representative samples can be generated from a generative model, such as GANs, rather than replaying the original samples which can be large in number. It is argued that GANs generate new samples which may not belong exactly to one of the classes, so a new generative model is proposed. Experimental results are appealing.

I have some basic concerns.
(1) First of all, the idea of generating new samples for replay is not motivated well enough; in the experiments, even the model replaying on original images takes memory in few hundred MBs.
(2) Second, why is there a need for generating new samples? Why not select a representative subset of the original set of samples. There are tons of methods to select samples informatively, within the paradigm of active learning and beyond.
(3) Why not use conditional-GANs for generating data points specific to a class?

Since autoencoders have a problem of generating blurry images, the proposed autoencoding model borrows ideas from neural style transfer algorithm. Specifically, besides the reconstruction loss, content loss is introduced utilizing the idea of content transfer from the neural style transfer algorithm. It seem to make sense in reference to Fig. 2, though it needs better explanation.

For memory efficiency of the autoencoder itself, it is stores centroids and covariances of the episodes (representations of images), from which pseudo-encoded episodes are generated. This doesn't seem very principled, and may or may not work in different empirical settings, depending upon the intrinsic dimensionality of images, and the difficulty of the task.

---

> ### Author Response · Authors · 2020-11-14
> **Motivation and concerns addressed - More exerpiments added**
>
> We thank you for your insightful comments and will use these comments to improve the paper.
>
> Basic Concerns:
> 1) Idea of generating new samples for replay is not motivated well enough:
> As described in previous works (Rebuffi et al. 2017, Castro et al. 2018), one of the main problems faced by continual learning systems is to learn new tasks for a long period of time while using limited memory. Our paper, similar to previous works (iCaRL, EEIL etc.), perform experiments on benchmark datasets for a proof of concept and to have a fair comparison with previous works in continual learning. Because of the limited sizes of the datasets (and some datasets with low resolution images), even storing the complete datasets require memory in few hundred MBs. However, in real-world applications, as the number of new tasks continue to increase, the memory required to store the previous data will go out of bounds, since many real-world systems have limited on-board memory which can be in the range of a few GBs. As an example, we have added a new experiment in the paper (Section 4.5), in which we perform the continual learning experiment on ImageNet-50 dataset with images of size 224$\times$224. The total memory required to store all 65000 images belonging to only 50 classes requires 13.04 GB, which can cause a huge memory strain on the system. As the number of classes increase, this memory can easily go out of bounds. However, EEC only requires a maximum of 111.56 MB for the 50 classes (Section 4.5), which shows the significance of our approach to cope with limited memory.
> 2) why is there a need for generating new samples?:
> We agree that there are some techniques proposed to get a representative set of original samples. Some continual learning approaches, such as iCaRL and EEIL have used these techniques to store a representative set of original samples. However, as shown in the experimental results (Table 1), the approaches that store some original images of the old classes produce significantly lower accuracy compared to EEC and they start to suffer from catastrophic forgetting in the later increments. Simply storing a representative set of original images does not solve the catastrophic forgetting problem and leads to further issues like privacy and security, as argued by (Ostapenko et al. 2019) and in our paper (Section 1).
> 3) Why not use conditional-GANs (CGAN) for generating data points specific to a class?:
> We agree that using CGANs is a way to solve the problem of generating images belonging to no classes. However, the generated images are still not perfect. We have performed an experiment with CGAN on ImageNet-50 dataset and added it in the ablation study in the paper (Section 4.4). Results in Table 2 show that CGAN does not solve the problem of generating perfect images. Further, note that CGAN produces similar accuracy to other autoencoders that do not use the content loss, which shows that CGAN is not even an improvement over autoencoders without the content loss, even though these autoencoders produces blurry images (Figure 2).
>
> Pseudorehearsal is not principled enough:
> Pseudo-rehearsal is an intuitive and empirical technique inspired by the learning models in the brain (Robins, 1995), which has been used by previous works like FearNet (Kemker \& Kanan 2018). Because of lack of mathematical explanation, we have provided an empirical experiment to visually show how our technique works (Figures 3 and 5). This empirical experiment (Figure 3 and 5) and experiments with limited memory in Section 4.3 show the effectiveness of pseudorehearsal.
>
> Pseudorehearsal may or may not work depending upon the intrinsic dimensionality of images, and the difficulty of the task:
> Note that the dimensionality of the images does not have an effect on pseudorehearsal, since it is applied on the feature vectors produced by the autoencoders (see Section 4.5 for details). Also, we have tested our continual learning approach on the classification task with multiple datasets of varying difficulty levels and the empirical evaluations show that pseudorehearsal produced favorable results.

---

### Comment · ~Lucas_Caccia1 · 2021-01-18
**Some Related Works missing**

Hi,

Congratulations on your paper acceptance!

I would like to point out two papers which I think are most relevant to your proposed approach

(1) Scalable Recollections for Continual Lifelong Learning (AAAI 2019), where the authors trains an autoencoder with binary latent representations to reduce the storage requirements for experience replay https://ojs.aaai.org//index.php/AAAI/article/view/3935

(2) Online Learned Continual Compression with Adaptive Quantization Modules (ICML 2020). Disclaimer, I am an author of the paper . We show that training vector quantized autoencoder avoids the blurriness issues typical to VAEs, and that we can perform memory efficient replay for continual learning with little degradation in image quality. (http://proceedings.mlr.press/v119/caccia20a.html)

Please consider adding these works in the list or prior work.

Thank you

---

### Decision · Program_Chairs · 2021-01-07
**Final Decision**

**Decision:**

Accept (Poster)

**Comment:**

This paper uses an autoencoder with neural style transfer to generate images from previously seen classes to avoid catastrophic forgetting in continual learning.

While reviewers had some concerns about the paper (experiments on high-resolution images, comparison with FearNet), authors have addressed all the concerns. R1's concern about the motivation for generation instead of replaying actual images is not necessary since this is not the first work to use generative replay.